# Generation of High-Yield, Functional Oligodendrocytes from a c-myc Immortalized Neural Cell Line, Endowed with Staminal Properties

**DOI:** 10.3390/ijms22031124

**Published:** 2021-01-23

**Authors:** Mafalda Giovanna Reccia, Floriana Volpicelli, Eirkiur Benedikz, Åsa Fex Svenningsen, Luca Colucci-D’Amato

**Affiliations:** 1Department of Environmental, Biological and Pharmaceutical Sciences and Technologies, University of Campania “Luigi Vanvitelli”, 81100 Caserta, Italy; mafaldareccia@yahoo.it; 2Department of Pharmacy, School of Medicine and Surgery, University of Naples Federico II, 80131 Naples, Italy; floriana.volpicelli@unina.it; 3Faculty of Health Sciences, J.B. Winsløwsvej 21, 5000 Odense, Denmark; eib@health.sdu.dk; 4Department of Molecular Medicine, University of Southern Denmark, J. B. Winsløws Vej 21.1, 5000 Odense, Denmark; 5Interuniversity Center for Research in Neuroscience (CIRN), University of Campania “Luigi Vanvitelli”, 80131 Naples, Italy

**Keywords:** neural stem cell, oligodendrocyte, neuron, myelin, immortalization, cell culture

## Abstract

Neural stem cells represent a powerful tool to study molecules involved in pathophysiology of Nervous System and to discover new drugs. Although they can be cultured and expanded in vitro as a primary culture, their use is hampered by their heterogeneity and by the cost and time needed for their preparation. Here we report that mes-c-myc A1 cells (A1), a neural cell line, is endowed with staminal properties. Undifferentiated/proliferating and differentiated/non-proliferating A1 cells are able to generate neurospheres (Ns) in which gene expression parallels the original differentiation status. In fact, Ns derived from undifferentiated A1 cells express higher levels of Nestin, Kruppel-like factor 4 (Klf4) and glial fibrillary protein (GFAP), markers of stemness, while those obtained from differentiated A1 cells show higher levels of the neuronal marker beta III tubulin. Interestingly, Ns differentiation, by Epidermal Growth Factors (EGF) and Fibroblast Growth Factor 2 (bFGF) withdrawal, generates oligodendrocytes at high-yield as shown by the expression of markers, Galactosylceramidase (Gal-C) Neuron-Glial antigen 2 (NG2), Receptor-Interacting Protein (RIP) and Myelin Basic Protein (MBP). Finally, upon co-culture, Ns-A1-derived oligodendrocytes cause a redistribution of contactin-associated protein (Caspr/paranodin) protein on neuronal cells, as primary oligodendrocytes cultures, suggesting that they are able to form compact myelin. Thus, Ns-A1-derived oligodendrocytes may represent a time-saving and low-cost tool to study the pathophysiology of oligodendrocytes and to test new drugs.

## 1. Introduction

Neural Stem Cells (NSCs) are undifferentiated cells endowed with self-renewal capacities and multipotent differentiation into the three major cell types of the Central Nervous System (CNS), such as neurons, astrocytes and oligodendrocytes. NSCs play a critical role in the CNS development, homeostasis and repair and can be isolated and propagated from embryonic, neonatal and adult mammalian CNS. NSCs can be cultured and expanded as neurospheres, spherical cell clusters formed in a serum free medium supplemented with EGF and/or FGF2 in a non-adhesive substrate. Upon mitogenic growth factors removal and adhesion to the substrate, NSCs undergo differentiation into neurons, astrocytes and oligodendrocytes [1,2,3]. NSC manipulation is a target of intense investigation either for therapeutic purposes or to better understand mechanisms underlying brain functions [4,5,6]. Nevertheless, a number of important drawbacks, including slow growth kinetics and experimental variability, limits their use as a tool for basic research and/or for novel therapeutic approaches. Indeed, a number of experimental approaches such as high-throughput screening of new drugs or cell-based therapies require large amounts of cells and rapid, reproducible and low-cost procedures which NSC do not guarantee. A way to overcome these limits is to obtain a homogeneous, unlimited and mitotically competent source of neural progenitors and/or stem cells by means of their immortalization [7]. Several non-oncogenic immortalizing genes have been exploited to obtain clonal progenitor/stem cell lines. Among these genes, myc has often been used to induce immortalization of both non-neural and neural cells [7,8,9,10,11]. Myc codes for a transcription factor that acts as a regulator in metabolism, cell cycle, differentiation, cell lifespan and cell size control [12,13,14,15,16,17]. Interestingly, myc has been identified as a regulator of pluripotency, and used in the reprogramming of differentiated cells back to a pluripotent state [18]. Neural progenitors or neural stem cells immortalized with myc, display different plasticity that might depend on the myc mutant used for the immortalization (i.e., v-myc or c-myc T58A) and/or on the primary cells where was it introduced [7,8,19]. It is of note that, primary NSC/progenitor cells generally display a low propensity to generate oligodendrocytes, thus, making it difficult to study oligodendrocyte biology and/or to be used for therapeutic purposes. For reasons currently unknown, myc-immortalized NSCs/progenitor cells generate oligodendrocytes at much higher percentage than parental NSCs/progenitor cells [7,20]. In particular, upon growth factors withdrawal, human NSCs immortalized with a mutant c-myc give rise up to 25% oligodendrocytes differently from their parental cells capable to generate only 1–4% oligodendrocytes of the total cells [8].

In this study, we report the staminal properties of a neural cell line previously immortalized by means of c-myc infection of a ventral mesencephalic embryonic mouse culture (mes-c-myc A1cells) [21]. A1 cells have the ability to form neurospheres upon administration of bFGF and EGF, where the cells express both neural staminal markers as Nestin, Kfl4 and Sox2 as well as markers of the three neural cell lineages such as β III tubulin, GFAP and GalC. Upon growth factors withdrawal, the NSCs originating from A1 cells, can generate large amounts of bona fide oligodendrocytes, as shown by markers such as NG2, RIP and MBP. Moreover, we found that, similarly to primary oligodendrocytes, in co-culture experiments, A1 cells-derived oligodendrocytes are able to interact functionally with neuronal cells, as assessed by the typical caspr/paranodin redistribution on neurites [22]. Thus, our findings make A1 cells a useful tool to generate stem cells with a propensity to differentiate into large numbers of oligodendrocytes enabling the study of cellular and molecular aspects of brain pathophysiology and particularly molecules and signaling pathways of demyelinating diseases.

## 2. Results

### 2.1. Expression of Staminal and Neural Markers by und-Ns and diff-Ns

To investigate the effects of differentiation on gene expression, A1 cells were deprived of serum for 6 days and cAMP was added to the culture medium (Figure 1A, upper panel). As shown in the upper panel of Figure 1B, the differentiated A1 cells showed decreased mRNA levels of staminal markers such as Nestin, Klf4 and Sox2, as well as in astroglial and oligodendroglial markers (GFAP and GalC mRNA levels, respectively). The β III tubulin mRNA level was instead increased compared to undifferentiated cells.

After 5–7 days in culture in the presence of 20 ng/mL EGF, 10 ng/mL bFGF and N2 supplement (neurosphere culture medium), both undifferentiated and differentiated A1 cells formed neurospheres (Figure 1A, lower panel), which will be referred to as undifferentiated A1 cells-derived neurospheres (und-Ns) and differentiated A1 cells-derived neurospheres (diff-Ns), respectively. Und-Ns and diff-Ns showed statistically significant differences in gene expression of staminal and neural markers when analyzed by real-time PCR (Figure 1B, lower panel). mRNA levels of the staminal markers Nestin and Klf4, and the astroglial marker GFAP were higher in und-Ns and, conversely, the early neuronal marker β III tubulin showed a higher expression in diff-Ns. No significant difference was observed in the expression of Sox2 and GalC, staminal and oligodendroglial markers, respectively. The expression of staminal and neural markers in und-Ns was also investigated by immunofluorescence performed 24 h after dissociation and plating of the Ns-A1 cells on poly-D-lysine cover glasses (Figure 1C). A quantitative analysis of cells positive for each of the selected markers, revealed that most of the cells expressed the staminal marker Nestin (80.32 ± 8.86) and the immature oligodendroglial marker NG2 (67.40 ± 5.06), whereas 33.45 ± 3.79 of cells were positive for β III tubulin (Figure 1C). 

### 2.2. Molecular Characterization of und-Ns Subcultures

To study the staminal properties of A1 neurospheres, we subjected und-Ns to sequential passaging. Primary neurospheres, dissociated and cultured in neurosphere culture medium again, developed into secondary spheres, which, in turn, formed tertiary spheres (Figure 2A). To assess whether long-term A1 neurosphere culture affected staminal and neural gene expression, we studied the mRNA expression of the staminal markers Nestin, Klf4 and Sox2, the early neuronal marker β III tubulin, the astroglial marker GFAP and the immature oligodendroglial marker GalC in primary, secondary and tertiary A1 neurosphere cultures. As shown in Figure 2B Nestin, Klf4 and Sox2 mRNA levels were upregulated in tertiary neurospheres as compared to primary or secondary spheres, while β III tubulin and GalC transcription decreased in tertiary spheres as compared to primary or secondary spheres. GFAP expression showed no significant difference in the neurospheres subjected to sequential passages. These data suggested an increased proliferative capacity and a reduced expression of neuronal and oligodendroglial mRNAs in long-term passaged A1 neurosphere cultures. This finding is consistent with data reported in the literature showing that secondary and tertiary neurospheres tend to decrease neuronal and astroglial markers, whereas staminal markers such as Nestin and Sox2 are augmented [23,24,25].

### 2.3. Differentiation and Characterization of Primary Neurosphere Derived from A1 Undifferentiated Cells (Ns)

#### 2.3.1. Characterization of Neurosphere-Derived A1 Cells by Real-Time PCR

To characterize primary neurosphere derived from undifferentiated A1 cells, Ns were differentiated in the absence of mitogenic factors (as described in the materials and methods section, Figure 3A). Different time points were selected to characterize molecularly diff-Ns by real-time PCR (0, 3, 10 and 13 days) and immunofluorescence analysis (0, 3, 10, 17 and 24 days). We performed real-time PCR to analyze the expression of the following markers: the staminal marker Klf4, the neuronal marker β III tubulin, the immature oligodendroglial marker GalC, and the astroglial marker GFAP. As shown in Figure 3B Klf4, β III tubulin, GalC and GFAP mRNA levels decreased at 3, 10, 13 or 17 days of differentiation compared to Ns. The expression of the oligodendroglial and the neural markers decreased by passing from 3 to 10 days of differentiation. Interestingly, GalC mRNA levels decreased at 10 days of differentiation but increased again after further 3 days.

A summary of the gene expression of staminal and neural markers, analyzed by real-time PCR in A1 cells cultured in different differentiation conditions is reported in Table 1.

#### 2.3.2. Characterization of diff-Ns by Immunofluorescence

To better characterize protein expression in diff-Ns cells, immunochemistry was performed. The expression of oligodendroglial markers, i.e., NG2, expressed by oligodendrocyte precursor cells, RIP, a marker for both immature and mature oligodendrocytes, MBP, exclusively expressed in mature oligodendrocytes was analyzed along with the expression of NeuN and GFAP neuronal and astroglial markers, respectively. As shown in Figure 4 and in Appendix A, diff-Ns cells expressed early and late oligodendroglial markers. On day 0. 72% ± 5.4% of cells in culture expressed NG2. The NG2 expression decreased to reach 18.48% ± 5.55% on day 24. At the same time point, 53.7% ± 6.08% of cells were positive to RIP, that clearly increased during differentiation. Finally, the mature oligodendroglial marker MBP was expressed by 68.09% ± 6.08% of cells at the day 24 of differentiation.

To investigate if diff-Ns cells also express GFAP and NeuN, NG2-positive cells were double immunolabeled with NeuN while RIP-positive cells were double immunolabeled with GFAP (Figure 5, Appendix A). The late neuronal marker NeuN was expressed by about 10–20% of the cells at all days of differentiation, except on day 3 when 32.89% ± 4.2% of cells were positive for this marker. The expression of GFAP was limited to 7% ± 3.67% at day 3, and no GFAP positive cells were observed at the subsequent days of differentiation.

As shown in Figure 4 and Figure 4 at days 0 and 3 of differentiation, NG2-, RIP- and MBP-positive cells branch short and multiple processes from a round and small cell body (arrows), whereas they extend longer slim branches from a flat and larger cell body at the late days of differentiation (arrows), this morphological differentiation was coupled to a decrease in NG2 expression and to an increase in RIP and MBP expression.

### 2.4. Neuronal Differentiated A1 Cells When Co-Cultured With Differentiated Ns-A1 Express Oligodendroglial Markers, Redistribute Caspr/Paranodin to Paranodal Regions

Contactin-associated protein (caspr)/paranodin, whose expression is restricted to neurons, is uniformly/diffusely expressed on the surface of unmyelinated neurons whereas upon myelination it becomes redistributed from the internodes to the paranodal regions. Caspr expression and localization is regulated by glial ensheathment becoming concentrated to the paranods of mature myelinated fibers [22]. To assess functional properties of A1-derived oligodendrocytes, we searched for their ability to cause a redistribution of caspr immunoreactivity on neuronal cells upon co-culture. To this aim, we performed a co-culture experiment where A1-derived oligodendrocytes were directly added to neuronal differentiated A1 cells (Figure 6A). We first checked the expression of oligodendroglial and neuronal markers in this experimental setting (Figure 6A). Thus, after 24, 48 and 72 h, immunofluorescence was performed in co-cultures previously incubated with the following primary antibodies NG2, RIP and MAP2. Results showed that in co-culture, A1-derived oligodendrocytes still preserved the expression of the oligodendroglial markers NG2 and RIP and their processes frequently contacted MAP2-positive neuronal cells (Figure 6B). As also shown in Figure 6B, no RIP positive cells were detected when A1 cells differentiated towards a neuronal phenotype were cultured alone (Figure 6B).

Then, as shown in Figure 7A, a co-culture, containing A1-derived oligodendrocytes, differentiated for 24 days and A1 cells at 10 days of neuronal differentiation, was established. Double-immunolabeling with RIP and Caspr/paranodin was carried out. These co-cultures revealed a discontinuous RIP and Caspr-staining when neuronal differentiated A1 cells were closely associated with RIP-positive cells. Arrows reported in Figure 7C point to Caspr-positive regions interspersed with RIP-staining. This specific Caspr/paranodin and RIP localization is consistent with the formation of nodes of Ranvier, in which Caspr/paranodin is expressed at paranodal regions (Figure 7C), which are the site of closest apposition between the membranes of the axon and myelinating glia [26] and are regularly spaced by myelinated internodes. At paranodal regions the transmembrane protein Caspr/paranodin, located on the axolemma, binds the glial component neurofascin-155 [22,27,28]. Neuronally differentiated A1 cells, not interacting with RIP-positive cells in co-cultures, expressed Caspr/paranodin along their entire surface and processes (Figure 7B).

Finally, in order to corroborate our conclusion, a co-culture containing primary neurons and oligodendrocytes was prepared to compare its pattern of Caspr/paranodin staining with the one observed in the previous experiment (Appendix A). To do this a co-culture, containing cerebellar neurons and oligodendrocytes isolated from fetal rat CNS, was established. As shown in Appendix A, the immunofluorescence, using Caspr/paranodin and RIP antibodies, revealed a staining clustered in discrete areas, highly resembling the staining pattern observed in neuronal cells that interact with A1-derived oligodendrocytes. Thus, these data were highly suggestive that A1-derived oligodendrocytes are able to form compact myelin.

## 3. Discussion

In this study we reported that the A1 mes c-myc cell line had staminal properties and could, upon differentiation, generate a high yield of oligodendrocytes displaying functional properties.

NSCs are powerful tools in biomedical research to understand pathophysiological mechanisms involved in neural functions and diseases and to discover new compounds able to interfere with molecular signaling cascades. Although it is possible to generate NSCs from embryonic and even an adult brain, some drawbacks such as reproducibility, time and expensive experimental procedures, hamper their use as a biomedical tool. Therefore, established cell lines with staminal properties are a useful tool [29,30,31]. A1 cells were derived by the immortalization of a primary culture obtained from mouse embryonic ventral mesencephalon, using a retrovirus containing the c-myc proto-oncogene. It has shown the coexistence of both neuronal and glial properties [21]. When serum has been withdrawn and cAMP administered, A1 cells exit the cell cycle and undergo neuronal differentiation displaying long neurite processes sprouting from a round and refractive body. The generated neurons also have voltage-gated sodium and potassium channels that respond to depolarizing stimuli. These cells have been extensively characterized using advanced proteomic and genetic techniques thus uncovering genes and molecules associated with proliferation and differentiation [32,33,34,35]. Moreover, A1 cells have been instrumental in a number of experimental settings to uncover how drugs affect the responsiveness to neural cells according to their differentiation status [36,37,38,39,40,41]. Here we show that when cultured in suspension, in the presence of EGF and bFGF, undifferentiated A1 cells give rise to neurospheres. Molecular analysis of Ns-A1-derived showed an expression of stem cell, neuronal, astrocytic and oligodendrocyte markers such as Nestin, Klf-4, Sox2, β-III tubulin, GFAP and GalC, respectively, similar to primary neural stem cells.

We were also able to generate neurospheres from diff-A1Ns under the same conditions used for generating neurospheres from undiff-A1Ns. We found that the neurospheres originated from undifferentiated or differentiated A1 cells express different levels of markers according to the differentiation status of the cells of origin. Therefore, despite immortalization, the NS parallel their original status of differentiation and eventually their plasticity is not the same. For example, the expression of Nestin is higher in und-Ns originated from undifferentiated A1 cells whereas β-III tubulin is more abundant in diff-Ns generated from differentiated A1 cells.

Moreover, Sox2 is strongly upregulated in diff-Ns as compared to A1 differentiated cells from which they were originated. Sox2, one of the Yamanaka genes involved in reprogramming differentiated cells into induced Pluripotent Stem cells (iPSCs), is also expressed in neural stem cells where it is required for their maintenance and for long term in vitro culture [42,43,44]. Interestingly, KLF4 plays a critical role in maintaining self-renewal of ESCs [45,46,47] and is also one of the original four factors that reprogram somatic cells into iPSCs [48]. KLF4 is expressed in NSCs but drastically is downregulated in differentiated neurons [49]. Similarly, also in A1 cells KlF4 is highly expressed in undifferentiated cells and is downregulated in differentiated A1. It belongs to a family whose members play important roles in development, differentiation and functions of neural cells [50,51]. It is worth noting that KLF9 and KLF6 are required for oligodendrocyte differentiation and CNS myelination [52,53].

When tested for self-renewal, Ns were able to form secondary and tertiary neurospheres. As expected, when subjected to sequential passaging, an enrichment in cells expressing stem cell markers (i.e., Nestin, Klf4 and Sox2) was observed in secondary and tertiary Ns, whereas neuronal (i.e., β III-tubulin), astroglial (i.e., GFAP) and oligodendroglial (i.e., GalC) markers were decreased. 

The enrichment and/or the reduction of cells expressing the above mentioned markers upon passaging, is reported in literature since passaging foster amplification of less differentiated cells [23,24,25].

A1 cells, both und and diff-Ns as well as their undifferentiated and differentiated counterparts, express the oligodendrocyte marker GalC, together with neuronal and astroglial markers, consistently with their stem cell properties. The galactocerobroside GalC is an oligodendrocyte marker widely found in stem cells [7]. Ns are also positive for NG2, a chondroitin sulfate proteoglycan that is a marker of oligodendrocyte progenitor cells (OPCs). Although NG2 expressing cells were reported to be multipotent in vitro, Kang et al [54] showed that in vivo, postnatal CNS were able to generate myelinating oligodendrocytes, but not neurons or astrocytes. Moreover, in the presence of neurodegeneration or demyelinization or upon brain injuries, NG2 cells proliferate and participate in cell replacement, displaying high degree of plasticity [55,56,57]. When subjected to a program of differentiation A1NS show a consistent increase of RIP and MBP positive cells whereas NG2 cells are strongly decreased. This is consistent with a differentiation pathway towards a mature oligodendrocyte phenotype. In particular, the RIP antibody identifies 2′,3′-Cyclic-nucleotide 3′-phosphodiesterase (CNPase), an enzyme specifically localized in the cytoplasm of oligodendrocytes and in myelin where it represents about the 4% of the total myelin proteins [57]. Interestingly, CNPase that is fairly specific for myelin, is also a marker of myelin destruction in a number of CNS diseases, including MS, being detected in cerebrospinal and body fluids [58]. Along with CNPase, MBP positive cells are also consistently augmented during differentiation. MBP is a marker of mature and myelinated oligodendrocytes [59]. It is worth noting that upon growth factor withdrawal A1 Ns give rise to a majority of bona fide oligodendroglial cells (up to 70%). Although, it is known that, differently from primary cultures, established immortalized cell lines show a much higher proclivity to generate oligodendrocytes, to the best of our knowledge, such a percentage has never been observed [7,8,20].

Finally, the experiment of co-culture clearly shows that Ns-A1-derived oligodendrocytes cause a redistribution of Caspr/paranodin protein on neuronal cells, as in the case of primary oligodendrocytes, strongly suggesting that they are able to form compact myelin. Although this kind of experiment is widely used to show functional oligodendrocytes, it is worth to mention that transmission electronic analysis would provide a more conclusive data about the formation of myelin [60]. To generate myelinating oligodendrocytes in vitro is crucial because myelination is essential for axonal integrity. Oligodendrocytes and axons are completely interdependent functional units and dysfunctions in one result in a loss of function of the other. Recent data indicate that oligodendrocytes metabolically support the axon and express a wide range of neurotransmitter receptors and ion channels playing key roles in neural circuit plasticity, learning and the long-term integrity of axons and neurons [61]. To have an in vitro system useful for investigating disruption or loss of capacity to generate myelinating oligodendrocytes, could facilitate understanding of the devastating effects on CNS function and neurodegenerative disease. Ns-A1-derived oligodendrocytes may represent a time-saving and low-cost tool to study the pathophysiology of oligodendrocytes and to test new drugs.

## 4. Materials and Methods

### 4.1. A1 Cell Culture Conditions

The immortalized A1 cell line was generated from mesencephalic primary cultures from 11-day-old CD1 (Charles River Laboratories, Milan, Italy) mouse embryos [21]. A1 cells cultured in MEM/F12 (ThermoFisher Scientific, Milan, Italy) supplemented with 10% fetal bovine serum (FBS) (ThermoFisher Scientific) proliferate actively and differentiate toward a neuronal phenotype after serum withdrawal and 1 mM cAMP (Sigma Aldrich, Milan, Italy) and N2 supplement (ThermoFisher Scientific) addition.

Undifferentiated and differentiated A1 cells cultured at the density of 2,5 × 10^5^/mL in MEM-F12 serum free medium, in the presence of 20 ng/mL of EGF (Sigma Aldrich), 10 ng/mL of bFGF (R&D System, Milan, Italy) and N2 supplement (ThermoFisher Scientific) are able to form neurospheres. Of neurosphere culture medium 10 mL was used for 25 cm^2^ flasks with no substrate pretreatment. A1 neurospheres are formed in 5–7 days when incubated at 37 °C in a humidified incubator with 5% CO_2_.

To obtain A1 neurosphere subcultures, the medium with suspended spheres between 150–200 microns was centrifuged at 700 rpm for 5 min at room temperature, the supernatant was discarded and the spheres were mechanically dissociated into a single cell suspension pipetting up and down through the tip of the 200 μL automatic pipette. The presence of single cells was confirmed by microscopy and their viability by trypan blue was assessed at the Burker chamber. Single cell suspension derived from dissociated primary neurospheres was cultured in neurosphere culture medium in suspension in 25 cm^2^ flask again. Secondary spheres were formed in 5–7 days when incubated at 37 °C in a humidified incubator with 5% CO_2_ [33]. The demonstration of self-renewal of NSCs requires more than one or two passages, thus A1 neurospheres were cultured for more than 8 weeks.

### 4.2. Neurosphere-Derived A1 Cell Differentiation

To induce neurosphere-derived A1 cell differentiation, individual spheres were mechanically dissociated into a single cell suspension as previously reported and plated on poly-D-lysine (Sigma Aldrich) precoated plates in neurosphere culture medium. At 24 h after plating, a time point referred to as day 0 (d0), EGF (Sigma Aldrich) was removed and cells were cultured without EGF (Sigma Aldrich) for 3 days (d3) to allow the generation of neural progenitors. Subsequently also bFGF (R&D System) was removed and the medium, without growth factors, was replaced every 3 days until 24 days. RNA isolation and immunostaining were performed to identify the phenotype of Ns cells (see Section 4.5 and Section 4.6).

### 4.3. Co-Culture of Neuronal Differentiated A1 Cells and Differentiated Ns-A1 Expressing Oligodendroglial Markers

Diff-Ns cells expressing oligodendroglial markers were co-cultured with A1 differentiated cells expressing neuronal markers. Briefly, 50,000 Ns cells, grown up in culture for 24 days (at this time points they most prominently expressed mature oligodendroglial markers) were directly added to 4 well plates containing 250,000 A1 cells plated on poly-D-lysin (1 mg/mL, Sigma Aldrich)) cover glasses and differentiated towards a neuronal phenotype supplementing the medium with cAMP (Sigma Aldrich) for 10 days. Co-cultures were maintained in N2 media for 24, 48 or 72 h. In the absence of cAMP (Sigma Aldrich), A1 cells differentiated to neurons for 10 days, retain a neuronal differentiation. Indeed, as previously described, A1 cells deprived of serum arrest cell division and shift toward a morphological neuron phenotype [21].

### 4.4. Primary CNS Cell Cultures

The Sprague-Dawley rats, housed at the central animal facility, University of Southern Denmark, were euthanized by carbon dioxide inhalation in accordance to the Danish and European legislation by authorized staff. Approval of the protocol was not required, because housing and handling procedures, and euthanasia, are well-standardized protocols conceived and constantly monitored by the Animal Research Ethics Committee, Denmark. The rats were euthanized for the specific purposes of this study and no experimental procedures were performed on the rats prior to euthanasia. Primary embryonic CNS cultures were generated using modification of the published method [62]. Cerebellar cell cultures were prepared from embryos of Sprague-Dawley rats on gestation day 17. Briefly, pregnant female rats were euthanized with CO_2_ and subjected to a cesarean section. The embryos were removed and decapitated. The brains were dissected and then the cerebellum removed and transferred to a new dish with L-15 medium. The tissue was mechanically dissociated and passed through a cell strainer to remove cellular debris and centrifuged at 250 *g*. The cells were re-dissociated in neurobasal cell culturing medium supplemented with 2% B27, and 0.3% glutamine (all from ThermoFisher Scientific, Sweden). The cells were plated at a cell density of 1 × 10^6^ cells/mL and plated, on poly-l-lysine–coated permanox Lab-Tec 4 well chamber slides (Nalge Nunc International). The cell cultures were grown for 4 weeks and then immunolabeled using the antibody RIP (Developmental Studies Hybridoma Bank, University of Iowa, Iowa City, IA, USA), and Caspr 1 (a kind gift from the lab of Dr. David R. Colmans lab, McGill University, Montreal, QC, Canada).

### 4.5. RNA Extraction, Reverse Transcription and Real-Time PCR Analysis

Isolation of the total RNA from cells was performed by using the TRIzol reagent (ThermoFisher Scientific) according to the manufacturer’s instructions. Total cellular RNA (2 μg) was reverse transcribed using high-capacity cDNA reverse transcription kits (Applied Biosystems, Carlsbad, CA, USA). Samples containing water were included as controls. cDNA was then used for real-time PCR. Real-time PCR was performed in an ABI PRISM^®^ 7500 Sequence Detection System (Applied Biosystems) using SYBR Green PCR Master Mix 5× HOT TAQ EvaGreen qPCR Mix Plus (Microtech Research Products, Napoli, Italy). The reaction mixture contained SYBR Green PCR Master Mix 1×, 50 ng of cDNA template and 300 nM of each forward and reverse primers in a final volume of 20 μL. The amplification conditions were as follows: 1 initial cycle for 15 min at 95 °C to activate hot Taq DNA polymerase, followed by 40 cycles for 15 s at 95 °C and 35 s at 63 °C, and a final extension for 35 s at 72 °C, and then 1 cycle of dissociation for 15 s at 95 °C, 1 min at 60 °C and 15 s at 95 °C. Control reactions included water. The primers used in the present study were listed in Table 2.

Each sample was tested in triplicate, and data were expressed as the quantity of specific transcripts subtracted from the quantity of the control gene hypoxanthine-guanine phosphorybosiltransferase (HPRT). ΔΔC_T_ values for use in fold change calculation were quantified by the following formula:ΔΔC_T_ = (C_T_ TARGET − C_T_ HPRT)_UND/und-Ns/Ns_ − (C_T_ TARGET − C_T_ HPRT)_DIFF/diff-Ns/d3,d10,d17_(1)

Fold change was determined using 2^−ΔΔCT^ formula and, the standard deviation (S.D.) was calculated based on the final fold change values.

### 4.6. Immunocytochemistry

Cells were seeded on poly-D-lysin (1 mg/mL, Sigma Aldrich) precoated coverslips, washed with PBS 1 × 3 times (5 min each) and fixed in 4% paraformaldehyde in PBS for 10 min at room temperature. After fixation, cultures were incubated with a solution containing 0,25% TritonX-100 and 0,25% BSA for 1 h at room temperature to permeabilize and block aspecific binding sites. The following primary antibodies were incubated overnight at 4 °C: β III tubulin/Tuj1 (monoclonal antibody MMS-435P, 1:500, Covance, UK), GFAP (polyclonal antibody, 1:250, Sigma Aldrich), Nestin (monoclonal antibody, 1:200, Abcam, Cambridge, UK), NG2 (polyclonal antibody, 1:500, Millipore), RIP (monoclonal antibody, 1:500, ThermoFisher Scientific), MBP (polyclonal antibody, 1:500, ThermoFisher Scientific), neuronal nuclei (NeuN) (monoclonal antibody, 1:500, ThermoFisher Scientific), micro-tubule-associated protein 2 (MAP2) (polyclonal antibody, 1:500, Abcam) and Caspr/paranodin (monoclonal antibody, 1:500, Abcam). After three washes in PBS 1× (5 min each), following secondary antibodies were applied for 1 h: anti-rabbit Alexa488 (1:500; Molecular Probes, ThermoFisher Scientific), anti-rabbit Rhodamine Red-X (1:600, Jackson ImmunoResearch), anti-mouse Alexa546 (1:500; Molecular Probes, ThermoFisher Scientific), anti-mouse Alexa488 (1:500; Molecular Probes, ThermoFisher Scientific) and anti-chicken Alexa647 (1:400, Molecular Probes, ThermoFisher Scientific). Nuclei were counterstained with 0.1 μg/mL of 4′,6-diamidino-2-phenylindole (DAPI). The nuclei staining was performed for 5 min at RT. After washing in PBS, cells were mounted in DABCO (Sigma Aldrich) and viewed on a Leica epifluorescence microscope at 20× and 63× magnification.

Primary antibodies were omitted as a negative control to assess the specificity of immunostaining. For double immunofluorescence, cells were incubated overnight in a mixture of two primary antibodies harvested in different species. To analyze staminal neural markers expression, ten random fields containing at least 100 cells, were captured and analyzed using Image J software. The percentage of positive cells was calculated by counting the number of positive cells as a proportion of blue DAPI positive nuclei.

### 4.7. Statistical Analysis

Results are expressed as means ± S.D. Statistical analysis was achieved by a two-tailed *t*-test (Microsoft Excel software). Results of the analysis with a value *p* < 0.01 and *p* < 0.05 were considered to be statistically significant (** *p* < 0.01, * *p* < 0.05).

## Figures and Tables

**Figure 1 ijms-22-01124-f001:**
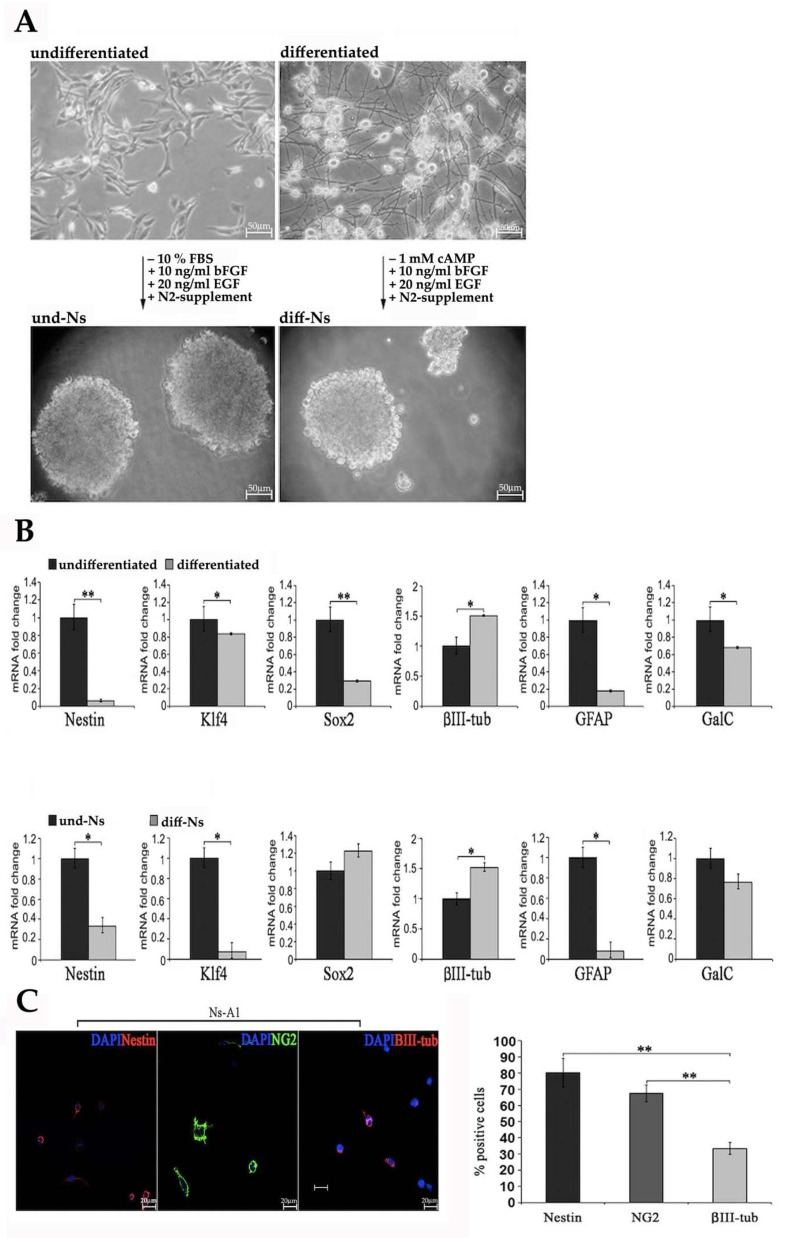
Expression of staminal and neural markers by und-Ns and diff-Ns. (**A**) (upper panel) Undifferentiated (und) and differentiated (diff) A1 cells deprived of FBS and cAMP, respectively and cultured in a medium supplemented with bFGF, EGF and N2 form (black arrows) neurospheres (Ns): und-Ns and diff-Ns, respectively (lower panel). Scale bar = 50 µm. (**B**) Expression of the staminal markers Nestin, Klf4 and Sox2 and neural markers βIII-tubulin, GFAP and GalC, investigated by real-time PCR in undifferentiated and differentiated A1 cells (upper panel), und-Ns and diff-Ns (lower panel). Data expressed as mRNA fold changes with respect to und (upper panel) or und-Ns A1(lower panel) are reported as mean ± S.D. ** *p* < 0.01 und versus diff A1 cells; * *p* < 0.05 und versus diff A1 cells; * *p* < 0.05 und-Ns versus diff-Ns. (**C**) Immunofluorescence analysis of Nestin (red), NG2 (green), βIII-tub (red) and DAPI nuclei (blue) in neurosphere-derived A1 (Ns-A1) cells cultured in presence of mitogenic factors for 24 h (day 0). Scale bar = 20 µm. The percentage of positive cells was calculated by counting the number of Nestin-, NG2- and βIII-tubulin-positive cells as a proportion of blue DAPI positive nuclei.

**Figure 2 ijms-22-01124-f002:**
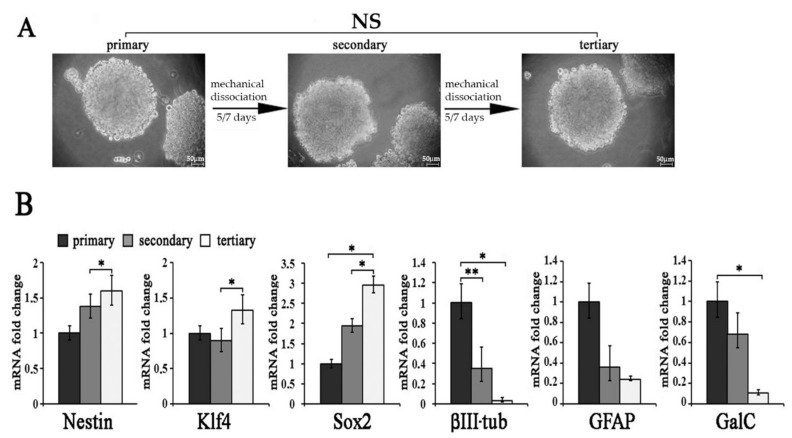
Molecular characterization of und-Ns subcultures (**A**) Neurospheres (Ns) originated from undifferentiated A1 cells (und-Ns) could self-renew when mechanically dissociated. Primary neurospheres give rise to secondary spheres, which in turn form tertiary spheres when sequentially passaged. Scale bar = 50 µm. (**B**) Nestin, Klf4, Sox2, β III tubulin, GFAP and GalC transcription, analyzed by real-time PCR, in primary, secondary and tertiary neurospheres. Data expressed as mRNA fold changes with respect to primary spheres are reported as ± S.D. ** *p* < 0.01 primary versus secondary Ns; * *p* < 0.05 primary versus tertiary Ns; * *p* < 0.05 secondary versus tertiary Ns.

**Figure 3 ijms-22-01124-f003:**
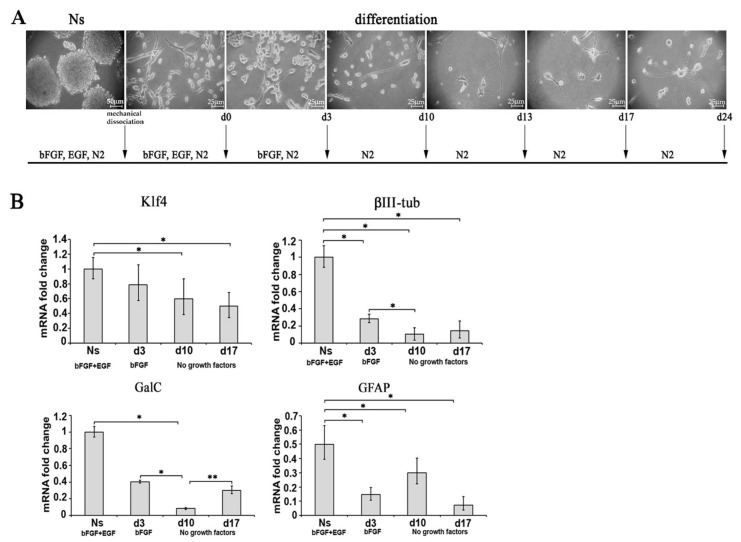
Characterization of diff-Ns cells by real-time PCR. (**A**) A1 cells derived from the mechanical dissociation of neurospheres (Ns) were plated on poly-D-lysine coated plates and differentiated by removing EGF at day 0 and bFGF at day 3. Cells were maintained in N2 media in the subsequent days of culture. Scale bars = 50 µm, 25 µm. (**B**) real-time PCR analysis of the expression of Klf4, β III tubulin, GalC, GFAP in Ns and diff-Ns cells at 3 (d3), 10 (d10) and 13 (d13) or 17 (d17) days of differentiation. Data, expressed as mRNA fold changes with respect to Ns are reported as ± SD. * *p* < 0.05 Ns versus d3, d10, d13 or d17; * *p* < 0.05 d3 versus d10; ** *p* < 0.01 d10 versus d13.

**Figure 4 ijms-22-01124-f004:**
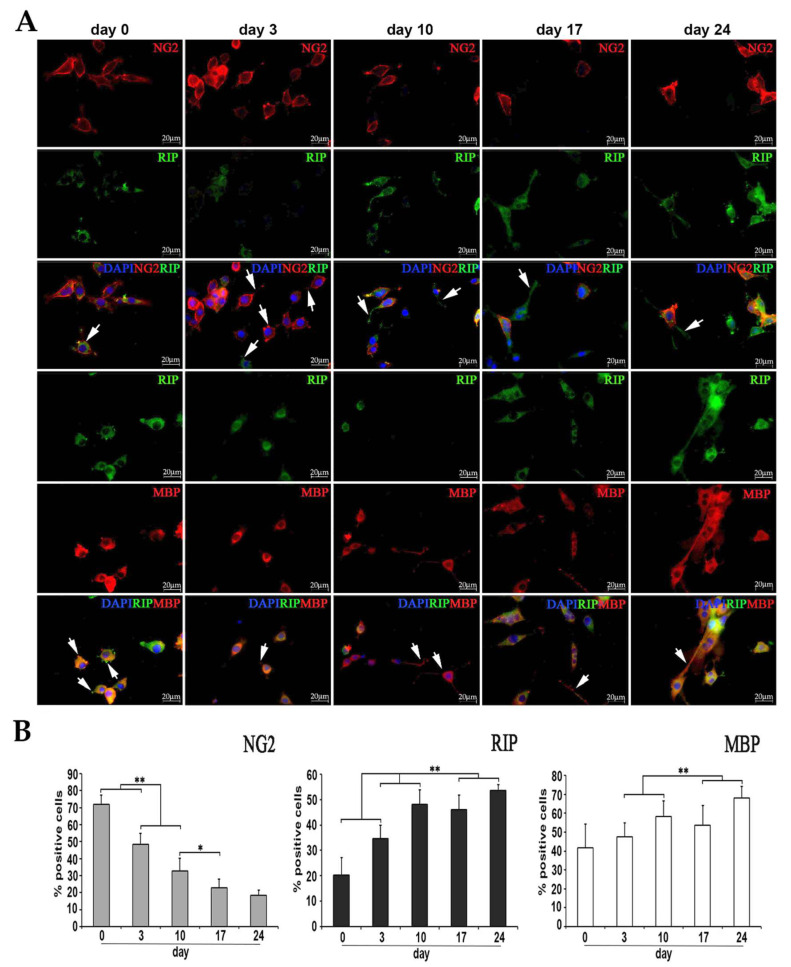
Characterization of diff-Ns by immunofluorescence. (**A**) Immunofluorescence analysis of the early oligodendroglial marker NG2 (red), and late oligodendroglial markers RIP (green), MBP (red) and DAPI nuclei (blue) in diff-Ns as described in Figure 3A. RIP positive cells were double-immunolabeled with NG2, for oligodendrocyte progenitors, or with MBP, for oligodendrocytes. (**B**) The percentage of positive cells was calculated by counting the number of NG2-, RIP- and MBP-positive cells as a proportion of blue DAPI positive nuclei. ** *p* < 0.01, * *p* < 0.05 day 3, 10, 17 or 24 versus the previous day of differentiation. Cells expressing oligodendroglial markers at the days 0 and 3 of differentiation branch short processes from a round and small body (arrows) and extend slender longer branches from a flattened and larger cell body at the late days of differentiation (arrows). Scale bar = 20 μm. Magnification, 63×.

**Figure 5 ijms-22-01124-f005:**
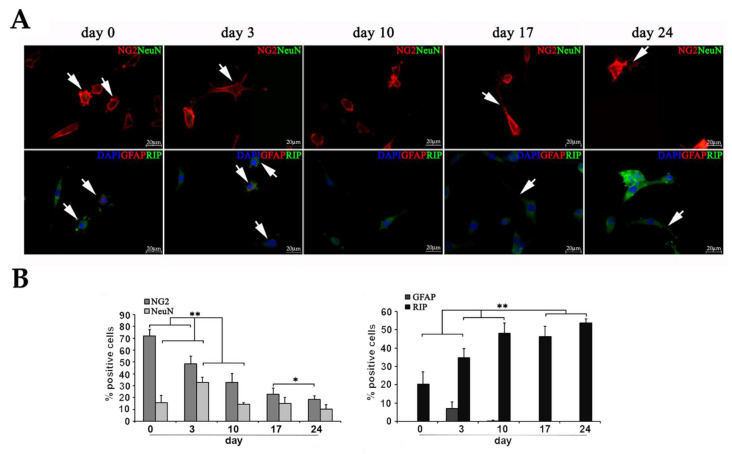
Characterization of diff-Ns by immunofluorescence. (**A**) Immunofluorescence analysis of GFAP (red), RIP (green), NG2 (red), NeuN (green) and DAPI nuclei (blue) in diff-Ns differentiated as described in Figure 3A. (**B**) The percentage of positive cells was quantified by counting the number of GFAP-, RIP-, NG2- and NeuN-positive cells as a proportion of blue DAPI positive nuclei. Cells expressing oligodendroglial markers at the days 0 and 3 of differentiation branch short processes from a round and small body (arrows) and extend slim longer branches from a flat and larger cell body at the late days of differentiation (arrows). ** *p* < 0.01, * *p* < 0.05 day 3, 10, 17 or 24 versus the previous time of differentiation. Scale bar = 20 μm. Magnification 63×.

**Figure 6 ijms-22-01124-f006:**
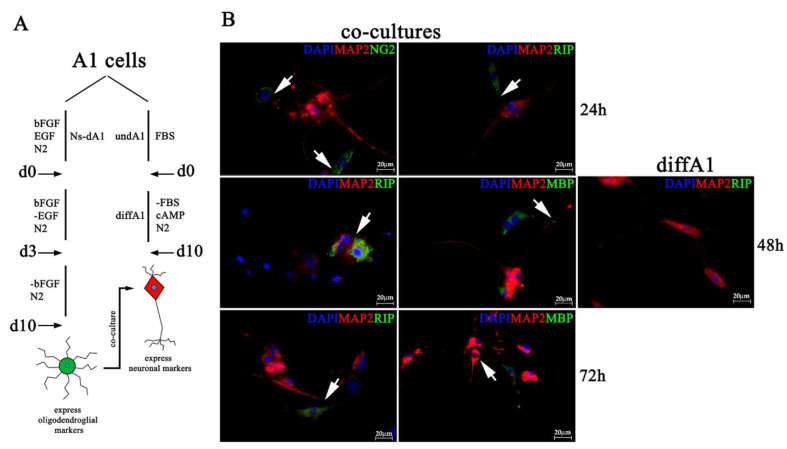
A1-derived oligodendrocytes preserved the expression of the oligodendroglial markers NG2 and RIP. (**A**) Ns cells differentiated for 10 days as shown in Figure 3A were directly added to A1 cells differentiated towards a neuronal phenotype for 10 days. (**B**) Co-cultures were immunolabeled with MAP2 (red), NG2 (green), RIP (green), MBP (green) and DAPI nuclei (blue) after 24, 48 and 72 h. NG2 and RIP-positive cells extended processes that frequently reach MAP2-positive cells (arrows). No RIP-positive cells were found in A1 cells differentiated (diffA1) towards a neuronal phenotype. Scale bar = 20 μm. Magnification 63×.

**Figure 7 ijms-22-01124-f007:**
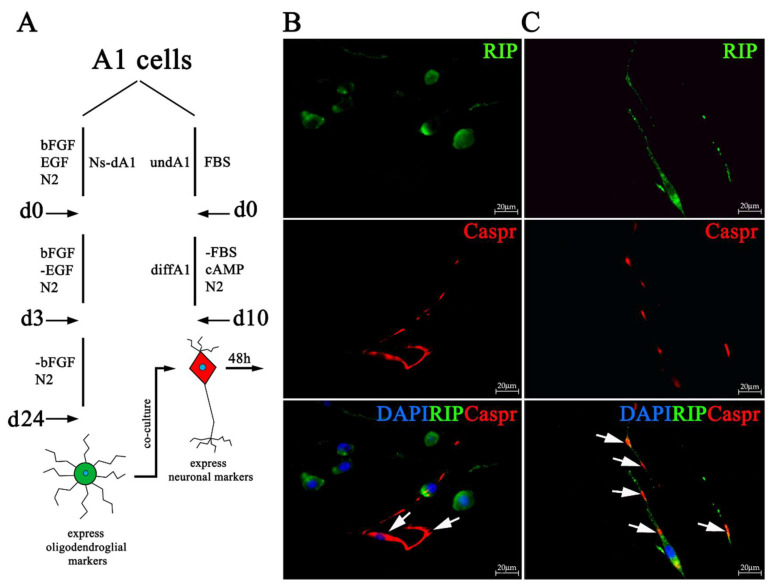
(**A**) Ns cells differentiated for 24 days as shown in Figure 3A were directly added to A1 cells differentiated towards a neuronal phenotype for 10 days. (**B**,**C**) Immunofluorescence analysis of Caspr (red), RIP (green) immunoreactive cells and DAPI nuclei (blue) in co-cultures immunolabeled after 48 h. (**B**) Neuronal differentiated A1 cells not interacting with RIP-positive cells expressing Caspr/paranodin along their entire surface and processes (arrows), (**C**) whereas a discontinuous RIP and Caspr-staining is observed in co-cultures when RIP-positive cells are closely associated neuronal differentiated A1 cells (arrows). Scale bar = 20 μm. Magnification 63×.

**Table 1 ijms-22-01124-t001:** Summary of gene expression of staminal and neural markers analyzed by real-time PCR in A1 cells cultured in different condition. The table compares undifferentiated (und) and differentiated (diff) A1 cells, neurospheres derived from undifferentiated (und-Ns) and differentiated A1 cells (diff-Ns). Neurosphere-derived A1 (Ns-A1) cells were differentiated as described in Figure 3A at different days of differentiation. Genes highly expressed are denoted with ++, while those with medium positive expression with +, and no detectable expression are denoted with -, N.D. is used for the analysis not performed.

	und *vs.* diff		und-Ns *vs.* diff-Ns		diff Ns-A1 d3	diff Ns-A1 d10	diff Ns-A1 d13	diff Ns-A1 d17	diff Ns-A1 d24
**Klf4**	++	+	++	+	+	+	N.D.	+	N.D.
**GalC**	++	+	+	+	++	+	++	N.D.	N.D.
**ßIII-tub**	+	++	+	++	++	+	+	N.D.	N.D.
**GFAP**	++	+	++	+	-	-	N.D.	-	N.D.

**Table 2 ijms-22-01124-t002:** Primers (5′-3′) used for real time-PCR.

Gene of Interest	Primers Sequence
Nestin	Forward 5′- AGCAACTGGCACACCTCAAGA- 3′ Reverse 5′- CTCAGCCTCCAGCAGAGTCC- 3′
βIII-tubulin	Forward 5′- CGTGGGCTCAAAATGTCATC- 3′ Reverse 5’- TGGCTGTGAACTGCTCCGAGAT- 3’
GFAP	Forward 5′- GAGGGACAACTTTGCACAGGA- 3′ Reverse 5′- CCAGCCTCAGGTTGGTTTCAT- 3′
Sox2	Forward 5′- AGGGCTGGACTGCGAACTG- 3′ Reverse 5′- TTTGCACCCCTCCCAATTC- 3′
HPRT	Forward 5′- TGCCCTTGACTATAATGAGTACTTCAG- 3′ Reverse 5′- TTGGCTTTTCCAGTTTCACTAATG- 3′
GalC	Forward 5′- GCCTTTCATTCCAAATCCCAA- 3′ Reverse 5′- AGCTCACTGCCTACTATGTTG- 3′
Klf4	Forward 5′- CCCTCCTTTCCTGCCAGACC- 3′ Reverse 5′- ACCTTCTTCCCCTCTTTGGCTTG- 3′

## Data Availability

Not applicable.

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
