# Peer review of "Generation of High-Yield, Functional Oligodendrocytes from a c-myc Immortalized Neural Cell Line, Endowed with Staminal Properties"

_ijms, 2021, doi:10.3390/ijms22031124_

Round 1
Reviewer 1 Report
The MS entitled “Generation of High-Yield, Functional Oligodendrocytes From a c-myc Immortalized Neural Cell Line, Edowed With Staminal Properties” presents the characterization and differentiation of a previously generated mouse mesencephalic neural stem cell line immortalized with c-myc (named A1) and routinely cultured and proliferated in undefined conditions. The authors claim that this cell line can generate a homogeneous culture of oligodendrocyte, after proliferation and differentiation in defined conditions (proliferation with FGF2/EGF).
The claim is based on the analysis of MBP and Gal-C markers as counting the positive cells after 28 days of differentiation of A1 neural progenitor cells. However, the anaylsis of GalC was performed only up to day 13 and shows anyway a decrease, as compared with the starting progenitor population! (Fig 3 and table 1). Regarding the MBP expression, it is reported as cell counts in Fig 4. 40-50% of the initial cells (undifferentiated!) express MBP, and around 60% of cells express MBP at day 10, which is quite maintained, with a small increase but without statistical significance, at day 28 (68,09 ± 6,08%).
More than this claimed efficient differentiation, that is not proved, the other presented experiments show a variable composition of the undifferentiated A1 cells, strongly depending on the culture conditions and passage number (here presented as primary, secondary and tertiary neurosheres) (Fig 2). In addition, it is not clear which cell populations (Ns) were used at the beginning of differentiation toward oligodendrocytes (Fig 3).
In conclusion, this MS needs a better experimental design, in case this line can be directed to oligodendrocytes, which can be anyway better characterized. The presented stainings were not convincing, positive and negative control are needed.
Regarding the “staminal” story, again the results of the neural stem/progenitor cell profile in A1 cell line are unclear. What means the staminal markers nestin, Klf4 and Sox2? At Line 301-3012 “Differently from Sox2, Klf4, while expressed in differentiated A1 cells, is decreased when these cells are induced to become stem cells”. Which cells are induced to become stem cells? If not with a staining for the A1 cells, at least the expression of Klf4 in the cell populations in the embryonic mesencephalon should be better documented.
Author Response
Response to Reviewer 1 Comments
Please find below, in red, detailed point-by-point responses to the reviewers’ comments.
Point 1: The claim is based on the analysis of MBP and Gal-C markers as counting the positive cells after 28 days of differentiation of A1 neural progenitor cells. However, the anaylsis of GalC was performed only up to day 13 and shows anyway a decrease, as compared with the starting progenitor population! (Fig 3 and table 1). Regarding the MBP expression, it is reported as cell counts in Fig 4. 40-50% of the initial cells (undifferentiated!) express MBP, and around 60% of cells express MBP at day 10, which is quite maintained, with a small increase but without statistical significance, at day 28 (68,09 ± 6,08%).
Response 1: Our conclusions, are based upon experiments carried out using a panel of well-established markers for oligodendrocytes and oligodendrocytes precursors such as NG2, RIP and also GalC and MBP. Moreover, we have carried out these experiments with two different techniques such as the Real-time PCR and the immunofluorescence to detect RNA expression and the presence of the protein, respectively. We have not analyzed all the markers with both techniques, at all times points. However the pictures that emerges is a clear differentiation towards a mature oligodendrocyte phenotype. In particular, RIP (CNPase) a well-known marker of mature oligodendrocyte shows an increase from about 20-25% positive cells at day 0 up to about 60% positive cells at day 24, while immature oligodendrocyte, that is NG2-positive cells, at the same time points, decrease from about 70% to 20% (Fig4). In the same Fig.4, MBP-positive cells counted at day 3 (40-50%) and at day 4 (around 70%) significantly increase. The fact that at time 0, about 40-50% cells express MBP, corroborates the general picture that A1 cell-derived neurospheres are committed towards an oligodendrocyte phenotype. To put our results in the context of the literature, we would like to pinpoint that neurospheres generated from immortalized neural precursors when differentiated with a similar protocol give rise, for reasons currently unknown, to more oligodendrocytes than the non-immortalized NSC counterpart (see De Filippi et al PLoS One 2008, 3, e3310, doi:10.1371/journal.pone.0003310; De Filippi et al. Stem Cells 2007, 25, 2312-2321, doi:10.1634/stemcells.2007-0040. Villa et al Exp Cell Res 2009, 315, 1860-1874, doi:10.1016/j.yexcr.2009.03.011).
Although, as described in the above mentioned papers, the percentage of oligodendrocytes generated from neurospheres varies, according to the species, the immortalizing gene etc., it reaches a maximum of 23% (De Filippi et al PloSOne 2008). Thus 70% of oligodendrocytes obtained in our case is quite remarkable. Moreover, the experiment of co-culture (Fig.7) shows that Ns cells differentiated into oligodendrocyte when cultured with bona fide neuronal cells, cause a redistribution of Caspr/paranodin protein on neuronal cells, as in the case of primary oligodendrocytes, strongly suggesting that they are able to form compact myelin like functional oligodendrocytes.
On the whole, we think that the bulk of the experiments, the techniques used and the markers analyzed, allow us to claim that a differentiation towards a mature oligodendroglial phenotype occurs.
Just a last remark concerning these points. In the paper we have never shown a time point at day 28, as claimed by the reviewer. Likely, the reviewer was confused by the small size of characters and numbers. In this case, we apologize and, accordingly, we have enlarged them.
Point 2: More than this claimed efficient differentiation, that is not proved, the other presented experiments show a variable composition of the undifferentiated A1 cells, strongly depending on the culture conditions and passage number (here presented as primary, secondary and tertiary neurosheres) (Fig 2). In addition, it is not clear which cell populations (Ns) were used at the beginning of differentiation toward oligodendrocytes (Fig 3).
Response 2: The variable composition of undifferentiated A1 cells, (i.e. the coexistence of markers of different cell lineages such as staminal, neuronal, astroglial and oligodendroglial markers) is typical of cells endowed with neural stem cell properties (Cacci et al Exp Cell Res 2007, Kondo et al Proceedings of the National Academy of Sciences Jan 2004, 101 (3) 781-786; DOI:10.1073/pnas.0307618100).
Moreover, neurospheres formation is a further test to show that neural stem cells are present within a given cell culture. In this case, although the markers of all neural cells (i.e. neuron, astrocyte and oligodendrocyte) and of stemness are kept, it is expected that, upon changing the substrate and/or the condition of culture the frequency of markers change (Fig1). This is how neural stem cells behave in culture and such a behavior is well described in the literature (Gage FH. Mammalian neural stem cells. Science. 2000 Feb 25;287(5457):1433-8. doi: 10.1126/science.287.5457.1433. PMID: 10688783; Osterberg N, Roussa E. Characterization of primary neurospheres generated from mouse ventral rostral hindbrain. Cell Tissue Res. 2009;336(1):11-20. doi:10.1007/s00441-008-0743-0).
Similarly, in case of primary, secondary and tertiary neurospheres it is consistent with data reported in the literature that cell lineage markers change upon passages (Lindley RM, Hawcutt DB, Connell MG, Edgar DH, Kenny SE. Properties of secondary and tertiary human enteric nervous system neurospheres. J Pediatr Surg. 2009 Jun;44(6):1249-55; discussion 1255-6. doi: 10.1016/j.jpedsurg.2009.02.048. PMID: 19524749; Osterberg N, Roussa E. Characterization of primary neurospheres generated from mouse ventral rostral hindbrain. Cell Tissue Res. 2009;336(1):11-20. doi:10.1007/s00441-008-0743-0; Peng C, Lu L, Li Y, Hu J. Neurospheres Induced from Human Adipose-Derived Stem Cells as a New Source of Neural Progenitor Cells. Cell Transplant. 2019;28(1_suppl):66S-75S. doi:10.1177/0963689719888619).
However, we must not have been clear and convincing enough, thus, following the reviewer’s comments, we have added new references and changed the text accordingly in order to corroborate our results and to put them in the context of the literature (see line 155).
Concerning, “which cell populations (Ns) were used at the beginning of differentiation toward oligodendrocytes (Fig 3)”, we agree with the reviewer that in the text it was not clear. Indeed, we have used primary und-Ns, that is primary neurospheres originated from undifferentiated A1 cells. Thus, we changed the text accordingly (line 168 and 170).
Point 3: In conclusion, this MS needs a better experimental design, in case this line can be directed to oligodendrocytes, which can be anyway better characterized. The presented stainings were not convincing, positive and negative control are needed.
Response 3: The experimental design, such as the techniques used (i.e. qRT-PCR and IF), the cell markers that we analyzed (i.e. NG2, RIP, GalC, MBP, Nestin, Sox2, Klf4, GFAP, b III tubulin, NeuN and Craspr/paranodin), the time points (up to 24 days) are well-established and widely used in this type of experiments as shown by papers published in respected journals (De Filippi et al Stem cells 2007; De Filippi et al PloSOne 2008; Villa et al Exp Cell Res, 2009; Zhang Y, Lu XY, Casella G, et al. Generation of Oligodendrocyte Progenitor Cells From Mouse Bone Marrow Cells. Front Cell Neurosci. 2019;13:247. Published 2019 Jun 5. doi:10.3389/fncel.2019.00247).
Concerning the staining, as asked by the reviewer, we have included a new supplementary figure 2 showing some double immunofluorescence labelling (Abs: MBP, RIP, NG2, GFAP, NeuN) with a nuclear counterstain. In this case a lower magnification shows that not all cells are positive and thus it is unlikely that the antibodies are not specific (new supplementary Fig.2).
Finally, following the reviewer’s comments, in order to make the conclusions more consistent with the results we have slightly changed the Discussion (line 376) and we have also added a sentence and a new reference (line 383).
Point 4: Regarding the “staminal” story, again the results of the neural stem/progenitor cell profile in A1 cell line are unclear. What means the staminal markers nestin, Klf4 and Sox2? At Line 301-3012 “Differently from Sox2, Klf4, while expressed in differentiated A1 cells, is decreased when these cells are induced to become stem cells”. Which cells are induced to become stem cells? If not with a staining for the A1 cells, at least the expression of Klf4 in the cell populations in the embryonic mesencephalon should be better documented.
Response 4: Concerning Klf4, a well-established marker of stemness that was used in the paper to characterize cells, we agree with the reviewer that the sentence was not clear, therefore we have re-formulated the sentence and also added new references to better describe the expression of Klf4 (line 338-343).
We would like to thank the reviewer for his constructive criticisms and insightful comments that helped us to improve our review.

Reviewer 2 Report
The work of Reccia et al is devoted to the characterization of a new immortalized cell line suitable for reprogramming into a glial or neuronal phenotype. In general, adequate cellular models, including 3D-models, suitable for studying processes in neurodegenerative diseases, are extremely relevant and timely. The work was performed at a good methodological level, and the results obtained look reliable. Minor shortcomings of the work include too small print in Figures 1B, 3B, 4B, 5B. In addition, it is desirable to increase the contrast in fluorescent photographs to improve the clarity of the results obtained.
Author Response
Response to Reviewer 2 Comments
Please find below, in red, detailed point-by-point responses to the reviewers’ comments.
Point 1: Minor shortcomings of the work include too small print in Figures 1B, 3B, 4B, 5B.
Response 1: We replaced the small print characters in figures 1B, 3B, 4B, 5B.
Point 2: In addition, it is desirable to increase the contrast in fluorescent photographs to improve the clarity of the results obtained.
Response 2: We slightly increased the contrast in fluorescent photographs.
We would like to thank the reviewer for his constructive criticisms and insightful comments that helped us to improve our review.
